# A Prospective Birth Cohort Study on Maternal Cholesterol Levels and Offspring Attention Deficit Hyperactivity Disorder: New Insight on Sex Differences

**DOI:** 10.3390/brainsci8010003

**Published:** 2017-12-23

**Authors:** Yuelong Ji, Anne W. Riley, Li-Ching Lee, Heather Volk, Xiumei Hong, Guoying Wang, Rayris Angomas, Tom Stivers, Anastacia Wahl, Hongkai Ji, Tami R. Bartell, Irina Burd, David Paige, Margaret D. Fallin, Barry Zuckerman, Xiaobin Wang

**Affiliations:** 1Center on the Early Life Origins of Disease, Department of Population, Family and Reproductive Health, Johns Hopkins University Bloomberg School of Public Health, 615 N Wolfe St, Baltimore, MD 21205, USA; yji7@jhu.edu (Y.J.); ariley1@jhu.edu (A.W.R.); xhong3@jhu.edu (X.H.); gwang24@jhu.edu (G.W.); dpaige@jhu.edu (D.P.); 2Department of Epidemiology, Johns Hopkins University Bloomberg School of Public Health, 615 N Wolfe St, Baltimore, MD 21205, USA; llee38@jhu.edu (L.-C.L.); dfallin@jhu.edu (M.D.F.); 3Wendy Klag Center for Autism and Developmental Disabilities & Department of Mental Health, 615 N Wolfe St, Baltimore, MD 21205, USA; hvolk1@jhu.edu; 4Department of Pediatrics, Boston University School of Medicine and Boston Medical Center, 1 Boston Medical Center Place, Boston, MA 02118, USA; Rayris.Angomas@bmc.org (R.A.); Tom.Stivers@bmc.org (T.S.); awahl@bu.edu (A.W.); barryzuckerman1@gmail.com (B.Z.); 5Department of Biostatistics, Johns Hopkins University Bloomberg School of Public Health, 615 N Wolfe St, Baltimore, MD 21205, USA; hji@jhu.edu; 6Stanley Manne Children’s Research Institute, Mary Ann & J. Milburn Smith Child Health Research, Outreach and Advocacy Center, Ann & Robert H. Lurie Children’s Hospital of Chicago, 225 E Chicago Avenue, Chicago, IL 60611, USA; TBartell@luriechildrens.org; 7Integrated Research Center for Fetal Medicine, Department of Gynecology and Obstetrics, Johns Hopkins University School of Medicine, 1800 Orleans St, Baltimore, MD 21287, USA; iburd@jhmi.edu; 8Division of General Pediatrics & Adolescent Medicine, Department of Pediatrics, Johns Hopkins University School of Medicine, 1800 Orleans St, Baltimore, MD 21287, USA

**Keywords:** high-density lipoprotein, triglyceride, sex difference, ADHD

## Abstract

Growing evidence suggests that maternal cholesterol levels are important in the offspring’s brain growth and development. Previous studies on cholesterols and brain functions were mostly in adults. We sought to examine the prospective association between maternal cholesterol levels and the risk of attention deficit hyperactivity disorder (ADHD) in the offspring. We analyzed data from the Boston Birth Cohort, enrolled at birth and followed from birth up to age 15 years. The final analyses included 1479 mother-infant pairs: 303 children with ADHD, and 1176 neurotypical children without clinician-diagnosed neurodevelopmental disorders. The median age of the first diagnosis of ADHD was seven years. The multiple logistic regression results showed that a low maternal high-density lipoprotein level (≤60 mg/dL) was associated with an increased risk of ADHD, compared to a higher maternal high-density lipoprotein level, after adjusting for pertinent covariables. A “J” shaped relationship was observed between triglycerides and ADHD risk. The associations with ADHD for maternal high-density lipoprotein and triglycerides were more pronounced among boys. The findings based on this predominantly urban low-income minority birth cohort raise a new mechanistic perspective for understanding the origins of ADHD and the gender differences and future targets in the prevention of ADHD.

## 1. Introduction

In the US, attention deficit hyperactivity disorder (ADHD) is one of the most common neurodevelopmental disorders in children; its prevalence has risen from 7.0% to 10.2% among children aged 4–17 years during the past two decades [1], representing a nearly 5% increase each year since 2003 [2]. ADHD is characterized by inattention, hyperactivity, or impulsiveness [3,4,5], and is three times more common among males than females [6]. Approximately 66% to 85% of children diagnosed with ADHD will carry their disorder into adolescence and adulthood [7,8]. A 2007 estimation of the annual cost of ADHD in the US, including the cost of related health care utilization, medication, education, crime, and unemployment, was $14,500 per child ($42.5 billion in total) [9]. While ADHD medications have shown to be effective in controlling ADHD symptoms, they neither preclude the rising incidence of ADHD nor cure ADHD, not to mention that they are also the causes for additional costs and potential side effects [2]. Given its high prevalence and continuously rising trend, the impact of ADHD on individual families and society is expected to increase dramatically [7,9,10]. 

At present, our knowledge regarding the biological mechanisms of ADHD development and effective ways to prevent ADHD is insufficient. While research has identified several potential etiological mechanisms, such as gene variants, brain structural abnormalities, and neurotransmitter deficiency and dysregulation [11,12], much more work is needed to fully understand the early life determinants of ADHD and significant sex differences in ADHD risk. There is an urgent need to identify modifiable early life risk factors for ADHD, which are essential to the primary prevention efforts. Well-recognized environmental risk factors for ADHD include parent-related factors [13,14,15,16,17,18,19,20,21,22,23,24,25], low birthweight and preterm birth [26], exposure to organophosphates [27], polychlorinated biphenyls [28,29], and lead [28,30,31,32]. Besides those factors, multiple recent studies indicate that maternal metabolic profiles may also influence offspring’s neurodevelopment. For example, findings in the Boston Birth Cohort showed a strong association between maternal obesity and diabetes and increased risk of autism in childhood [33]. A large longitudinal study, using prospective pregnancy cohorts from the Nordic Network, showed that both overweight moms and moms with excessive weight gain during gestation had an over two-fold higher risk of having ADHD children [34]. However, no study has investigated the role of maternal dyslipidemia (a condition often associated with obesity or metabolic syndrome) in offspring’s ADHD development. 

Maternal cholesterol levels are biologically plausible to influence neurodevelopment in the offspring [33,34,35,36,37,38,39,40]. Besides cholesterol’s key functions, such as hormone synthesis, fat-soluble vitamin digestion and absorption, cell membrane stabilization, and inter-cellular communication, it is essential for normal brain development, especially during in-utero and early childhood [35,36,37]. Nearly 70% to 80% of brain cholesterol is present in myelin [41]. While fetal cholesterol can be synthesized endogenously [38], the placenta also delivers cholesterol from maternal circulation to the fetus through multiple cholesterol-carrying lipoproteins, such as low-density lipoproteins (LDL), high-density lipoproteins (HDL) and very low-density lipoproteins (VLDL) [39,40]. It was estimated that up to 20% of fetal cholesterol in the first trimester is derived from maternal cholesterol via the placenta [38]. 

During normal pregnancy in humans, maternal blood cholesterol levels increase with gestational age to meet the increasing demands of fetal growth and development, especially with regards to the fetal brain [42,43,44]. Conceivably, a dysregulation in the amount and the type of cholesterol during critical developmental windows could lead to suboptimal neurodevelopment, and subsequently, ADHD symptoms in childhood. However, this possibility remains to be explored. To our knowledge, existing cholesterol studies in humans have mainly focused on mental health outcomes in adults, in which HDL levels have been found to be associated with multiple cognitive impairments and neurodegenerative diseases [45,46,47]. In particular, there is a lack of prospective birth cohort study to investigate the inter-generational impact of cholesterol on ADHD. 

To fill in the aforementioned knowledge gaps, in this study, we sought to examine the prospective association between maternal cholesterol levels 24–72 h after delivery and the development of ADHD in the offspring using a longitudinal birth cohort design. Findings from such a study have important clinical and public health implications. The current clinical guidelines for optimal cholesterol levels have been set for non-pregnant women based on cardio-metabolic outcomes, aiming to control cholesterol levels. However, the requirements for optimal nutrition, including cholesterols, are higher during pregnancy due to the increasing demands of the uterus, placenta, and fetal growth. Furthermore, no guidelines for cholesterol levels have been established for pregnant women in the context of fetal brain growth and long-term neurodevelopmental outcomes.

## 2. Materials and Methods

### 2.1. Study Sample

The Boston Birth Cohort (BBC) has successfully recruited mother-infant pairs at birth; the participation rate has been >90% among eligible mothers approached by the research staff. Details of the recruitment of the BBC were published previously [48,49]. Eligible mothers were those who delivered a single live birth at Boston Medical Center (BMC). Pregnancies resulting from in vitro fertilization, multiple-gestation pregnancies, deliveries induced by maternal trauma, or newborns with substantial congenital disabilities were not eligible for enrollment. The Institutional Review Board (IRB) of the Boston University Medical Center and Johns Hopkins Bloomberg School of Public Health approved the BBC study. Informed consent was obtained from each participant under the IRB approved protocol (IRB No. 00003966).

Of enrolled mother-infant pairs at birth in the BBC, 3098 who continued to receive pediatric primary care at BMC were enrolled in a postnatal follow-up study [33,48,50]. Our study sample excluded participants who had missing maternal cholesterol measurements and key covariates. We further excluded children with physician-diagnosed neurodevelopmental disorders other than ADHD (Appendix A). Our final analyses consisted of 1479 mother-infant pairs, including 303 children with ADHD and 1176 neurotypical children (Figure 1). The maternal and child characteristics for participants excluded and included are compared in Appendix A.

### 2.2. Data Collection Procedures and Measures of Key Variables

Mother-infant pairs were enrolled 24 to 72 h after birth. After obtaining informed consent, face-to-face interviews using a standardized questionnaire were conducted to collect mothers’ reports on family socio-demographics, substance use, and other prenatal exposure information. The maternal and newborn medical records were extracted using a standardized abstraction form. Since 2003, electronic medical records (EMRs) have become part of routine clinical data collection for the BBC, including both well-child and specialty medical visits at BMC. For each primary care visit, the EMRs contain the primary and secondary diagnoses from the International Classification of Diseases, Ninth Revision (ICD-9) (before 1 October 2015) and ICD-10 (after 1 October 2015).

Maternal serum total cholesterol (TC), triglycerides (TG), and high-density lipoprotein (HDL) levels were measured using nonfasting blood samples obtained between 24 to 72 h after delivery. Serum low-density lipoprotein (LDL) levels were calculated using the Friedwald equation. The detailed measurement and calculation methods are described in our previous publication [51]. Of note, nonfasting samples primarily impact TC and TG levels, which may be higher than in a fasting state.

The “ADHD group” was defined as having any of the following clinician-diagnosed ICD-9 codes: [314.0 (Attention deficit disorder of childhood), 314.00 (Attention deficit disorder without mention of hyperactivity), 314.01 (Attention deficit disorder with hyperactivity), 314.1 (Hyperkinesis with developmental delay), 314.2 (Hyperkinetic conduct disorder), 314.8 (Other specified manifestations of hyperkinetic syndrome), and 314.9 (Unspecified hyperkinetic syndrome)], or any of the following ICD-10 codes: F90.0 (ADHD, predominantly inattentive type), F90.1 (ADHD, predominantly hyperactive type), F90.2 (ADHD, combined type), F90.8 (ADHD, other type), and F90.9 (ADHD, unspecified type), as documented in the child’s EMRs.

The “neurotypical (NT) group” was defined as not having any clinician diagnosis of autism spectrum disorder, ADHD, conduct disorders, developmental delays, intellectual disabilities, failure to thrive, or congenital anomalies. This definition was established by clinical experts and has been applied by multiple published papers [52,53]. The ICD-9 and ICD-10 codes for the diagnoses of these developmental disorders are listed in Appendix A.

### 2.3. Statistical Analysis

The characteristics of the study sample between the “ADHD” and the “NT” groups were examined by t-test for continuous variables and χ^2^ test for categorical variables. TC, HDL, LDL, and TG were further analyzed as categorical variables based on clinically-established cut-off points [54,55], in addition to quartiles and the linear trend test. The clinical cut-off point for low HDL for women is <50 mg/dL [55]. The clinical cut-off point for non-fasting high TG is ≥200 mg/dL [54]. The quartile cut-off points were: TC (<176 mg/dL, 176–214 mg/dL, 215–254 mg/dL, >254 mg/dL), TG (<135 mg/dL, 135–176 mg/dL, 177–232 mg/dL, >232 mg/dL), HDL (<50 mg/dL, 50–60 mg/dL, 61–73 mg/dL, >73 mg/dL), and LDL (<96 mg/dL, 96–121 mg/dL, 122–150 mg/dL, >150 mg/dL). Next, we conducted multiple logistic regression (MLR) to examine the association between TC, HDL, LDL, and TG and the risk of ADHD diagnosis, both categorically and continuously, adjusting for maternal age at delivery, maternal race/ethnicity, maternal education, smoking during pregnancy, intrauterine infection, parity, child’s sex, mode of delivery, preterm birth, and birthweight. The effect of the interaction between child’s sex and each type of lipid or lipoprotein level on the risk of ADHD was tested using MLR and adjusted for the same set of covariates. Similarly, the joint effect of the child’s sex with each type of lipid or lipoprotein on the risk of ADHD was tested using MLR and adjusted for the same set of covariates except for child’s sex. In the sensitivity analyses, stratified analysis by each major covariate was conducted for the association between maternal HDL and ADHD. Furthermore, we repeated the above analyses within two subsets. One subset only included specialist-diagnosed ADHD as cases, while the other subset only included the ADHD cases whose age of last ADHD diagnosis was six years or older. We also performed additional analyses for the association between maternal cholesterol profiles and the risk of other neurodevelopmental disorders other than ADHD. All analyses were performed using STATA^®^ version 14.0 software (Stata Corporation, College Station, TX, USA). 

## 3. Results

There were 303 children with a clinician diagnosis of ADHD. Of these, 214 were diagnosed by a developmental specialist and 89 by a general pediatrician. The median age at the first ADHD diagnosis was seven years. Table 1 presents the bivariate comparisons of maternal and child characteristics between the “ADHD” and “NT” groups. The mothers of children with an ADHD diagnosis were more likely to have below college degree education, ever smoke before or during pregnancy, C-section delivery, lower TC, lower HDL, and lower LDL, compared with the neurotypical group. The children with any ADHD diagnosis were more likely to be older, male, born prematurely, and have had low birthweight, compared with the neurotypical group. The comparison results of major characteristics between excluded and included samples indicate that the included sample had less exposure to multiple risk factors, such as smoking during pregnancy, C-section delivery, lower gestational age, and lower birthweight (Appendix A).

Table 2 shows the MLR results for the effects of TC, HDL, LDL, and TG on the risk of any ADHD diagnosis, after adjusting for pertinent covariates. HDL <50 mg/dL, indicating a moderate risk of heart disease, was not associated with an increased risk of ADHD diagnosis (Odds Ratio (OR) = 1.30, 95% Confidence Interval (CI) (0.96, 1.74)). When HDL levels were analyzed as quartiles, mothers with first or second quartile HDL levels showed similarly increased odds of having a child with any ADHD diagnosis compared to those with fourth quartile HDL levels (Q2 vs. Q4: OR = 1.42, 95% CI (0.96, 2.09); Q1 vs. Q4: OR = 1.54, 95% CI (1.04, 2.28)). Mothers with ≤median HDL levels had a 39% increased odds of having a child with any ADHD diagnosis as compared to mothers with >median HDL levels (OR = 1.39, 95% CI (1.06, 1.82)). When HDL was analyzed as a continuous variable, the average odds of having a child with any ADHD diagnosis dropped 19% for every 20 mg/dL increase in maternal HDL levels (OR = 0.81, 95% CI (0.69, 0.95)). 

For TG, the risk of ADHD diagnosis for the children whose maternal TG levels were ≥200 mg/dL (indicating marginal risk of heart disease) was not statistically significantly different to those children whose mothers had <200 mg/dL TG levels (OR = 1.26, 95% CI (0.94, 1.68)). Compared to mothers with second quartile TG levels, the mothers with first, third or fourth quartile TG levels had a 51% increased odds of having a child with any ADHD diagnosis (OR = 1.51, 95% CI (1.08, 2.10)), suggesting a “J” shaped association. 

When LDL was analyzed as a continuous variable, the average odds of having a child with any ADHD diagnosis dropped 7% for every 20 mg/dL increase in maternal LDL levels (OR = 0.93, 95% CI (0.87, 0.99)). The MLR results for maternal TC levels did not show any significant association with the child’s ADHD diagnosis. 

Table 3 shows the associations between maternal HDL levels and the risk of any ADHD diagnosis stratified by the child’s sex and the joint effect of maternal HDL levels and the child’s sex on ADHD risk. As expected, compared to girls, boys had a three times higher risk of ADHD (OR = 3.25, 95% CI (2.45, 4.30)). The joint effects of maternal HDL and sex showed that boys whose mothers had ≤median HDL levels had increased odds of having any ADHD diagnosis (OR = 4.25, 95% CI (2.88, 6.26)), compared to girls whose mothers had >median HDL levels. The interaction term between sex and HDL was not statistically significant (OR = 1.35, 95% CI (0.77, 2.37)). Appendix A shows the stratified analysis results for the association between maternal HDL and ADHD. The results indicate that, besides child’s sex, smoking during pregnancy, intrauterine infection, parity, mode of delivery, gestational age, and birthweight also influence the association between maternal HDL and ADHD. Higher maternal HDL was more likely associated with a reduced risk of ADHD in the following stratum: boy, never smoker during pregnancy, no intrauterine infection during pregnancy, multiparous, vaginal delivery, full term, and normal birthweight. 

Appendix A shows the sensitivity analysis results on the joint effect of maternal HDL and sex by comparing children with specialist ADHD diagnosis and neurotypical children, and the findings were similar. Appendix A shows the results of the sensitivity analyses on the joint effect of maternal HDL and sex by excluding the children whose age of last ADHD diagnosis was under six years, and the findings were also similar. 

Table 4 shows the association between maternal TG levels and the risk of any ADHD diagnosis, stratified by the child’s sex and the joint effect of maternal TG levels and the child’s sex. The joint effects results showed that boys whose mothers had first, third or fourth quartile TG levels had a 394% increased odds of having any ADHD diagnosis (OR = 4.94, 95% CI (2.84, 8.58)), as compared to girls whose mothers had second quartile TG levels. The interaction term between sex and TG was not statistically significant (OR = 1.03, 95% CI (0.51, 2.07)). Appendix A shows the results of the sensitivity analyses on the joint effect of maternal TG and sex by comparing children with specialist ADHD diagnosis and neurotypical children, and the findings were similar. Appendix A shows the results of the sensitivity analyses on the joint effect of maternal TG and sex by excluding the children whose age at the last ADHD diagnosis was under six years, and the findings were also similar. These joint effects across HDL, TG, and sex are further illustrated in Figure 2 using MLR estimation and adjusting for the same covariates. Appendix A shows the results of additional analyses on the association between maternal cholesterol profiles and the risk of other neurodevelopmental disorders other than ADHD, such that maternal HDL and TG levels were not associated with other neurodevelopmental disorders.

## 4. Discussion

Despite the notion that cholesterol is essential for brain health, few prospective birth cohort studies have examined the effect of maternal cholesterol on offspring’s neurodevelopment. In the BBC, we found a significant association between maternal cholesterol levels, particularly HDL and TG measured 24–72 h after delivery (a proxy of peripartum maternal cholesterol levels), and ADHD risk in offspring. In contrast, this association is not significant for the risk of other neurodevelopmental disorders. Furthermore, our study sheds new light on the ADHD sex difference by demonstrating that boys are more vulnerable than girls to suboptimal maternal cholesterol levels. 

Our study findings were further strengthened by several aspects of our study design. We used clinician diagnosis extracted from the EMRs to define ADHD cases. More than half of the children in the ADHD group had over three ADHD clinician diagnoses in their EMRs. Additionally, over 80% of ADHD cases in the BBC were diagnosed by a neurodevelopmental specialist, thus, with a much higher specificity and lesser probability of case misclassification. The results of our sensitivity analyses, which restricted ADHD cases to those with a neurobehavioral specialist diagnosis and excluded those with a diagnosis at age younger than six years showed similar effect sizes and levels of significance as for our major findings.

While we cannot make a causality inference, and although biological mechanisms underlying the maternal HDL and child ADHD association remain to be determined, our findings are biologically plausible and in alignment with previous research. The central nervous system (CNS) is insulated from the systemic circulation by the blood brain barrier (BBB). Cholesterol and its carrying lipoproteins in the CNS are mainly synthesized locally within the brain [56,57] while cholesterol carried by plasma lipoproteins cannot move freely across the BBB [37,58]. Most lipoproteins found in the brain are synthesized by glial cells and astrocytes [57]. Additionally, the apolipoprotein B–containing lipoproteins, such as LDL, VLDL, and chylomicron, cannot enter the brain via the BBB [57]. Nevertheless, studies have suggested that plasma-based cholesterol may still affect the integrity and function of neurons and myelin [36,57]. For instance, the discoidal apolipoprotein A-I-containing HDL particles may enter the brain through scavenger receptor class B type I (SR-BI)-mediated uptake and transcytosis [57,59]. Notably, apolipoprotein A-I, which is the major component of plasma HDL, cannot be synthesized in the CNS [60,61]. After entering the CNS, it can further collect phospholipids and unesterified cholesterol and undergo maturation into HDL-like lipoproteins in the brain [57]. In addition to small plasma HDL particles, the side-chain oxidized oxysterols, such as 27-hydroxycholesterol, can also cross the BBB [62]. Moreover, peripheral HDL, even without crossing the BBB, may still influence fetal brain development due to its potential protective effect on cerebrovascular endothelial cell function [63]. In sum, the available evidence supports our findings regarding the protective effect of higher maternal HDL levels against ADHD risk in offspring.

The mechanism underlying the actions of maternal TG appears to be different from that underlying HDL. TG cannot cross the BBB but can influence multiple hormonal transportations across the BBB. For example, TG can effectively inhibit leptin transport across the BBB [64]. Besides its beneficial role in reducing obesity risk, leptin is a multifunctional hormone that influences many brain functions including appetite, motivation, learning, memory, and cognition [65]. 

If further confirmed by future investigation, our findings may have important research, clinical and public health implications. First, our data suggest that pregnant women should maintain a relatively higher level of HDL to meet the need for rapid fetal brain development during pregnancy and to reduce ADHD risk; this is particularly important for male fetuses. Our data indicate that the current clinical cut-off point for HDL (>50 mg/dL) for nonpregnant women, as recommended by the American Heart Association for reducing the risk of heart disease [54,55], may not be adequate for protecting against ADHD in offspring; thus, a higher cut-off point (>60 mg/dL) may be needed for identifying the fetus at risk for future ADHD. Lipid screening is not currently part of prenatal care guidelines, but it is relatively inexpensive and easily measured. Low HDL is modifiable by dietary and lifestyle changes and is treatable with pharmaceuticals. 

The long-observed and striking sex difference in ADHD risk continues to be poorly understood. Our study revealed that the maternal HDL and TG effects on ADHD are most pronounced among boys. This sex differences in response to suboptimal nutritional status are also found in other chronic diseases. For example, both human and animal studies showed that male fetuses are more likely to develop hypertension in response to the mother’s unfavorable nutrition and metabolism status during pregnancy [66,67,68,69,70]. One potential explanation is that male fetuses are more vulnerable to suboptimal maternal nutrition due to their more rapid in-utero growth compared to females [66,67,68,69,70].

Our study had the following limitations. First, our study only included a single measurement of maternal cholesterol taken 24–72 h after delivery. Ideally, a serial collection of lipid levels throughout pregnancy would best inform our hypotheses. At best, our one-time measurement reflects maternal cholesterol levels during peripartum. Additionally, the change in lipid levels between 24 and 72 h after delivery might add another potential source of variability for lipid measurement. Second, our study used non-fasting blood samples. The values for TC and TG levels may have been inflated in non-fasting blood samples and thus may have biased our study results towards the null. Further studies using fasting blood samples should be conducted to provide a more precise assessment of optimal TG levels during pregnancy. Third, our study was conducted in a US urban, low-income, primarily minority setting; thus, this was a population at higher risk of exposure to other risk factors for ADHD. Our analyses adjusted for known risk factors of ADHD, but could not adjust for multiple parent-related factors identified in previous studies such as poor parenting [13,14], maltreatment [15], conflict/parent-child hostility [23], and severe early deprivation [24,25]. Although our study sample is not representative of the general US population, research in urban minority populations is limited, and our study findings help to fill in this important data gap. Finally, our adjustment for known risk factors did not include some post-natal factors that could be related to both maternal cholesterol levels and ADHD risk, such as the child’s lipid levels. Although it is beyond the scope of this report, a study of the joint effects of cholesterol with other components of metabolic syndrome such as obesity, diabetes, and hypertension, may help to provide greater understanding about the associations between the maternal metabolic constellation and child neurodevelopmental outcomes. A previously published study did show that diabetes could cause a low production of brain cholesterol and its precursors, which in turn could lead to disruptions in synaptic formation and function [71]. Although our study occurred during the transition of the American Psychiatric Association’s Diagnostic and Statistical Manual (DSM) from the IV to the V edition, the diagnosis of ADHD in children did not change appreciably [72]. Moreover, the DSM-V lists both ICD-9 and ICD-10 codes for transition purposes [73].

## 5. Conclusions

In this large, prospective, predominantly US urban, low income, minority birth cohort, we found that suboptimal maternal cholesterol levels, in particular low HDL, may increase the risk of ADHD in offspring. The male fetus appears to be particularly vulnerable to suboptimal maternal cholesterol levels. Our findings raise new hypotheses for understanding of origins of ADHD, gender differences and future targets in the prevention of ADHD, and warrant additional investigation. 

## Figures and Tables

**Figure 1 brainsci-08-00003-f001:**
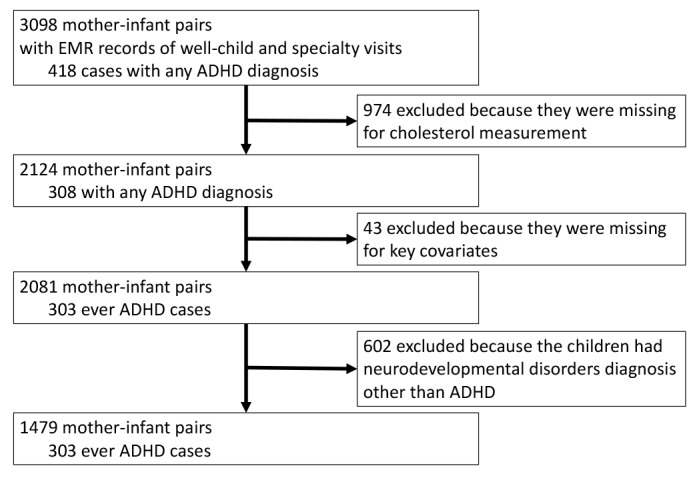
Flowchart of the sample included in the analyses.

**Figure 2 brainsci-08-00003-f002:**
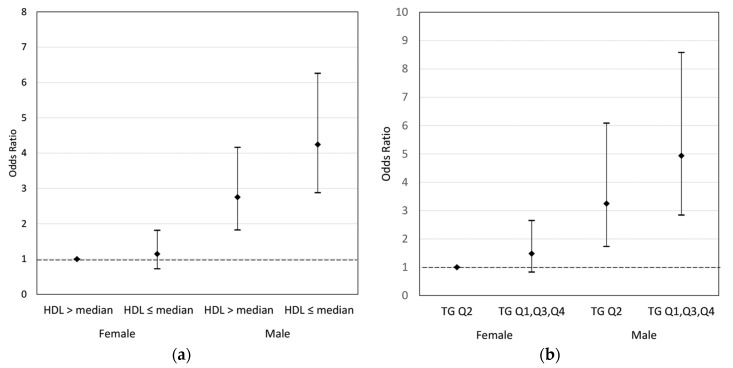
(**a**) The odds ratio of any ADHD diagnosis across maternal HDL and child’s sex groups using multiple logistic regression estimation; (**b**) the odds ratio of any ADHD diagnosis across maternal TG and child’s sex groups using multiple logistic regression estimation.

**Table 1 brainsci-08-00003-t001:** Maternal and child characteristics for children with any attention deficit hyperactivity disorder (ADHD) diagnosis and neurotypical children (NT).

Variable	Total, No. (%)	NT, No. (%)	ADHD, No. (%)	*p*-Value ^ǂ^
Total	1479 (100)	1176 (79.5)	303 (20.5)	
Maternal Age				0.317
<20	148 (10.0)	111 (9.4)	37 (12.2)	
20–34	1080 (73.0)	867 (73.8)	213 (70.3)	
≥35	251 (17.0)	198 (16.8)	53 (17.5)	
Education level				0.022
Below college degree	1278 (86.4)	1004 (85.4)	274 (90.4)	
College degree or above	201 (13.6)	172 (14.6)	29 (9.6)	
Race-ethnicity				0.230
Black	968 (65.5)	759 (64.5)	209 (69.0)	
White	74 (5.0)	56 (4.8)	18 (5.9)	
Hispanic	357 (24.1)	293 (24.9)	64 (21.1)	
Others	80 (5.4)	68 (5.8)	12 (4.0)	
Parity				0.901
Nulliparous	625 (42.3)	496 (42.2)	129 (42.6)	
Multiparous	854 (57.7)	680 (57.8)	174 (57.4)	
Smoking during pregnancy				<0.001
Never	1229 (83.1)	998 (84.9)	231 (76.2)	
Quitter	111 (7.5)	72 (6.1)	39 (12.9)	
Continuous	139 (9.4)	106 (9.0)	33 (10.9)	
Intrauterine infection				0.060
No	1292 (87.4)	1037 (88.2)	255 (84.2)	
Yes	187 (12.6)	139 (11.8)	48 (15.8)	
Child’s sex				<0.001
Female	749 (50.6)	664 (56.5)	85 (28.1)	
Male	730 (49.4)	512 (43.5)	218 (71.9)	
Delivery type				0.008
C-section	500 (33.8)	378 (32.1)	122 (40.3)	
Vaginal	979 (66.2)	798 (67.9)	181 (59.7)	
Season of child’s birth				0.797
Jan to March	333 (22.5)	264 (22.5)	69 (22.8)	
April to June	350 (23.7)	279 (23.7)	71 (23.4)	
July to September	402 (27.2)	314 (26.7)	88 (29.0)	
October to December	394 (26.6)	319 (27.1)	75 (24.8)	
Preterm birth (<37 weeks)				0.005
No	1125 (76.1)	913 (77.6)	212 (70.0)	
Yes	354 (23.9)	263 (22.4)	91 (30.0)	
Low birthweight (<2500 g)				0.028
No	1148 (77.6)	927 (78.8)	221 (72.9)	
Yes	331 (22.4)	249 (21.2)	82 (27.1)	
Gestational age, week				<0.001
Mean (SD)	38.1 (3.1)	38.2 (2.9)	37.5 (3.8)	
Birthweight, g				0.007
Mean (SD)	2996.7 (754.0)	3023.3 (716.4)	2893.5 (878.9)	
Maternal TC, mg/dL				0.018
Mean (SD)	219.6 (60.9)	221.5 (61.3)	212.2 (58.9)	
Maternal TG, mg/dL				0.838
Mean (SD)	191.9 (80.6)	192.2 (80.1)	191.1 (83.0)	
Maternal HDL, mg/dL				<0.001
Mean (SD)	62.0 (17.6)	62.8 (17.9)	58.8 (15.8)	
Maternal LDL, mg/dL				0.011
Mean (SD)	126.6 (41.8)	128.0 (42.1)	121.2 (39.9)	
Age of child, years				<0.001
Mean (SD)	10.6 (3.3)	10.3 (3.3)	11.7 (3.2)	

NT was defined as without any neurodevelopmental disorder diagnosis; ADHD was defined as any ADHD diagnosis; ^ǂ^ the *p*-values were obtained from chi-square tests or t-tests between children with and without any ADHD diagnosis.

**Table 2 brainsci-08-00003-t002:** The association between maternal cholesterol and the risk of ADHD in offspring.

Maternal Cholesterols		ADHD, No. (%)	NT, No. (%)	Crude OR	95% CI		*p*-Value	Adjusted OR	95% CI		*p*-Value
HDL clinical cut-off	≥50 mg/dL	213 (19.2)	898 (80.8)	1.00				1.00			
	<50 mg/dL	90 (24.5)	278 (75.5)	1.36	1.03	1.81	0.030	1.30	0.96	1.74	0.085
HDL quartiles	Q4 (>73 mg/dL)	55 (15.3)	304 (84.7)	1.00				1.00			
	Q3 (61–73 mg/dL)	67 (18.1)	304 (81.9)	1.22	0.82	1.80	0.322	1.11	0.74	1.67	0.606
	Q2 (50–60 mg/dL)	91 (23.9)	290 (76.1)	1.73	1.20	2.51	0.004	1.42	0.96	2.09	0.079
	Q1 (<50 mg/dL)	90 (24.5)	278 (75.5)	1.79	1.23	2.60	0.002	1.54	1.04	2.28	0.031
HDL binary	>median (60 mg/dL)	122 (16.7)	608 (83.3)	1.00				1.00			
	≤median (60 mg/dL)	181 (24.2)	568 (75.8)	1.59	1.23	2.05	<0.001	1.39	1.06	1.82	0.016
HDL linear trend (every 20 mg/dL increase)		303 (20.5)	1176 (79.5)	0.76	0.65	0.88	<0.001	0.81	0.69	0.95	0.011
TG clinical cut-off	<200 mg/dL	184 (19.8)	744 (80.2)	1.00				1.00			
	≥200 mg/dL	119 (21.6)	432 (78.4)	1.11	0.86	1.44	0.415	1.26	0.94	1.68	0.118
TG quartiles	Q1 (<135 mg/dL)	90 (23.9)	287 (76.1)	1.00				1.00			
	Q2 (135–176 mg/dL)	58 (16.3)	297 (83.7)	0.62	0.43	0.90	0.012	0.63	0.43	0.93	0.020
	Q3 (177–232 mg/dL)	76 (20.7)	291 (79.3)	0.83	0.59	1.18	0.300	0.88	0.61	1.27	0.495
	Q4 (>232 mg/dL)	79 (20.8)	301 (79.2)	0.84	0.59	1.18	0.309	0.98	0.66	1.44	0.909
TG binary	Q2	58 (16.3)	297 (83.7)	1.00				1.00			
	Q1, Q3, Q4	245 (21.8)	879 (78.2)	1.43	1.04	1.96	0.027	1.51	1.08	2.10	0.015
TG linear trend (every 20 mg/dL increase)		303 (20.5)	1176 (79.5)	1.00	0.97	1.03	0.838	1.02	0.98	1.06	0.348
LDL quartiles	Q1 (<96 mg/dL)	87 (23.6)	282 (76.4)	1.00				1.00			
	Q2 (96–121 mg/dL)	80 (21.8)	287 (78.2)	0.90	0.64	1.28	0.565	0.91	0.63	1.31	0.603
	Q3 (122–150 mg/dL)	67 (18.2)	301 (81.8)	0.72	0.50	1.03	0.074	0.82	0.57	1.20	0.316
	Q4 (>150 mg/dL)	69 (18.4)	306 (81.6)	0.73	0.51	1.04	0.083	0.76	0.52	1.11	0.153
LDL linear trend (every 20 mg/dL increase)		303 (20.5)	1176 (79.5)	0.92	0.87	0.98	0.011	0.93	0.87	0.99	0.033
TC quartiles	Q1 (<176 mg/dL)	92 (24.6)	282 (75.4)	1.00				1.00			
	Q2 (176–214 mg/dL)	73 (20.3)	287 (79.7)	0.78	0.55	1.10	0.161	0.82	0.57	1.18	0.289
	Q3 215–254 mg/dL)	72 (19.9)	290 (80.1)	0.76	0.54	1.08	0.125	0.86	0.59	1.25	0.424
	Q4 (>254 mg/dL)	66 (17.2)	317 (82.8)	0.64	0.45	0.91	0.013	0.73	0.50	1.08	0.111
TC linear trend (every 20 mg/dL increase)		303 (20.5)	1176 (79.5)	0.95	0.91	0.99	0.018	0.96	0.92	1.01	0.099

OR: Odds Ratio; CI: Confidence Interval; HDL: high-density lipoprotein; TG: triglycerides; LDL: low-density lipoprotein; TC: total cholesterol; NT was defined as without any neurodevelopmental disorder diagnosis; ADHD was defined as any ADHD diagnosis; the multiple logistic regression model was adjusted for maternal age at delivery, maternal race/ethnicity, maternal education, smoking during pregnancy, intrauterine infection, parity, child’s sex, mode of delivery, preterm birth, and birthweight.

**Table 3 brainsci-08-00003-t003:** The joint association of maternal high-density lipoprotein (HDL) levels and child’s sex with the risk of ADHD in offspring.

Sex	Maternal HDL	ADHD, No. (%)	NT, No. (%)	Adjusted OR	95% CI		*p*-Value
Female		85 (11.4)	664 (88.6)	1.00			
Male		218 (29.9)	512 (70.1)	3.25	2.45	4.30	<0.001
Joint effects of maternal HDL and sex							
Female	>median	42 (10.5)	359 (89.5)	1.00			
	≤median	43 (12.4)	305 (87.6)	1.14	0.72	1.81	0.564
Male	>median	80 (24.3)	249 (75.7)	2.75	1.82	4.16	<0.001
	≤median	138 (34.4)	263 (65.6)	4.25	2.88	6.26	<0.001

NT was defined as without any neurodevelopmental disorder diagnosis; ADHD was defined as any ADHD diagnosis; covariates included maternal age at delivery, maternal race/ethnicity, maternal education, smoking during pregnancy, intrauterine infection, parity, child’s sex, mode of delivery, preterm birth, and birthweight.

**Table 4 brainsci-08-00003-t004:** The joint association of maternal triglycerides (TG) levels and child’s sex with the risk of ADHD in offspring.

Sex	Maternal TG	ADHD, No. (%)	NT, No. (%)	Adjusted OR	95% CI		*p*-Value
Female		85 (11.4)	664 (88.6)	1.00			
Male		218 (29.9)	512 (70.1)	3.31	2.50	4.39	<0.001
Joint effects of maternal TG and sex							
Female	Q2	16 (8.8)	166 (91.2)	1.00			
	Q1, Q3, Q4	69 (12.2)	498 (87.8)	1.48	0.83	2.65	0.184
Male	Q2	42 (24.3)	131 (75.7)	3.25	1.73	6.09	<0.001
	Q1, Q3, Q4	176 (31.6)	381 (68.4)	4.94	2.84	8.58	<0.001

NT was defined as without any neurodevelopmental disorder diagnosis; ADHD was defined as any ADHD diagnosis; covariates included maternal age at delivery, maternal race/ethnicity, maternal education, smoking during pregnancy, intrauterine infection, parity, child’s sex, mode of delivery, preterm birth, and birthweight.

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
