# Peer review of "A Prospective Birth Cohort Study on Maternal Cholesterol Levels and Offspring Attention Deficit Hyperactivity Disorder: New Insight on Sex Differences"

_brainsci, 2017, doi:10.3390/brainsci8010003_

Round 1

Reviewer 1 Report

A Prospective Birth Cohort Study on Maternal 3 Cholesterol Levels and Offspring Attention Deficit 4 Hyperactivity Disorder: New Insight on Sex 5 Differences Yuelong Ji 1, Anne W. Riley 1, Li-Ching Lee 2,3, Heather Volk 3, Xiumei Hong 1, Guoying Wang 1 6 , Rayris Angomas 4, Tom Stivers 4, Anastacia Wahl 4, Hongkai Ji 5, Tami R. Bartell 6, Irina Burd 7 7 , David Paige 1, M. Daniele Fallin 2,3, Barry Zuckerman 4, Xiaobin Wang 8 1,8,*

This is an interesting and well-written paper reporting a study that found a significant association between maternal cholesterol levels, particularly HDL and TG measured 24-72 hours after delivery (a proxy of peripartum maternal cholesterol levels), and ADHD risk in offspring. Furthermore, the study sheds new light on the ADHD sex difference by demonstrating that boys are more vulnerable than girls to suboptimal maternal cholesterol levels.

The reported methods were appropriate to addressing the study questions. However, there are a few points that would serve to strengthen the paper should they be addressed.

1.       It makes sense to excluded participants who had missing maternal cholesterol measurements and key covariates. However, there might well have been important information to be gained had the study not excluded children with physician-diagnosed neurodevelopmental disorders other than ADHD (Table S1). 602 participants were excluded because the children had neurodevelopmental disorders diagnosis other than ADHD of which there were 303. The study was focusing on ADHD, but a persistent question in the literature pertaining to ADHD whether it be genetic, cognitive or metabolic is the nature of the risk factors that are specific to ADHD. Perhaps the authors are saving that data for a paper on other neurodevelopmental disorders. But that information would be very information if included in this paper.

2.       Could they specify the years during which participants were enrolled and the final year during which a participant could be classified as ADHD based on clinical diagnosis? Specifically, is it possible that some participants had not yet entered or passed through age of risk for ADHD. Rate of ADHD diagnosis would likely be considered low if 303 of 1479 children received an ADHD diagnosis (~2%). Age of children should be added to Table 1.

3.       89 only by a general pediatrician- suggest that word “only” be deleted from this sentence.

4.       Authors do an excellent job of discussing limitations in their study other than issues mentioned above. 

Author Response

Thank you so much for your helpful comments. Please check the PDF file for our response.

Reviewer 2 Report

This dense but well written paper explores the association between maternal cholesterol levels obtained within 72 hours post-partum and the subsequent risk of the child developing attention deficit hyperactivity disorder.   The underlining premise is that abnormal maternal cholesterol levels during critical periods of pregnancy may lead to suboptimal neurodevelopment in the foetus and subsequent ADHD symptoms in childhood. Data were derived from the Boston Birth Cohort.  The authors are constrained therefore by decisions about data collection that were not directed to the aims and objectives of the present study.  As such

Maternal cholesterol post-partum is only a proxy measure of cholesterol levels during pregnancy. Samples were not collected during the critical window for neurodevelopment in the foetus.

Cord blood would have provided a more direct indication of cholesterol levels within the newborn

Other considerations

The authors refer to a previous publication (reference 53) but paper 53 in the reference list has a completely different authorship and seems to relate to different topic. I did not check whether there were other inaccuracies in the referencing.

Blood samples for lipid analysis were taken between 24 and 72 hours post-partum. This would be reasonable if lipid levels remain stable through this period, but that may not be the case.  Bartels and O’Donoghue (Obstetric Medicine 2011; 4:147-151) claim that lipid levels fall back to close to prepregnancy levels within 24 to 72 hours post-partum. To be fair to the present authors however that statement seems to be based on a generalisation about levels measured sometime between 24 and 72 hours post-partum (Bartels et al. Journal of Obstetrics and Gynaecology 2012; 32:747-752)..  As far as I can tell a definitive study examining the trajectory of lipid levels between 24 and 72 hours postpartum has not been done. My aim point here, however, is that there is another potential source of variability (noise) in the data. 

I did not understand why the odds ratio for developing ADHD in males across HDL and TG groups was referenced to female favourable values for these variables. If the point is to demonstrate that male neurodevelopment and risk of ADHD is more sensitive to variability in HDL and TG levels than the comparison should be between favourable and unfavourable variables in males.   Based on visual inspection of figure 2a and 2b I think the difference is nonsignificant.

Author Response

(The authors gave the same response as above.)
